# Antifungal Activity of a Library of Aminothioxanthones

**DOI:** 10.3390/antibiotics11111488

**Published:** 2022-10-27

**Authors:** Joana Cardoso, Joana Freitas-Silva, Fernando Durães, Diogo Teixeira Carvalho, Luís Gales, Madalena Pinto, Emília Sousa, Eugénia Pinto

**Affiliations:** 1Laboratory de Organic and Pharmaceutical Chemistry, Department of Chemical Sciences, Faculty of Pharmacy, University of Porto, 4050-313 Porto, Portugal; 2Laboratory of Microbiology, Department of Biological Sciences, Faculty of Pharmacy, University of Porto, 4050-313 Porto, Portugal; 3Interdisciplinary Centre of Marine and Environmental Research (CIIMAR), University of Porto, 4450-208 Matosinhos, Portugal; 4Laboratory of Research in Pharmaceutical Chemistry, Department of Food and Drugs, Faculty of Pharmaceutical Sciences, Federal University of Alfenas, Alfenas 37137-001, Brazil; 5Institute of Biomedical Sciences Abel Salazar (ICBAS), University of Porto, 4050-313 Porto, Portugal; 6Institute of Molecular and Cellular Biology (i3S-IBMC), University of Porto, 4200-135 Porto, Portugal

**Keywords:** aminotioxanthones, antifungal activity, structure-activity relationship, virulence factors, mechanism of action

## Abstract

Fungal infections are one of the main causes of mortality and morbidity worldwide and taking into account the increasing incidence of strains resistant to classical antifungal drugs, the development of new agents has become an urgent clinical need. Considering that thioxanthones are bioisosteres of xanthones with known anti-infective actions, their scaffolds were selected for this study. A small library of synthesized aminothioxanthones (**1**–**10**) was evaluated for in vitro antifungal activity against *Candida albicans*, *Aspergillus fumigatus*, and *Trichophyton rubrum*; for the active compounds, the spectrum was further extended to other clinically relevant pathogenic fungi. The results showed that only compounds **1**, **8**, and **9** exhibited inhibitory and broad-spectrum antifungal effects. Given the greater antifungal potential presented, compound **1** was the subject of further investigations to study its anti-virulence activity and in an attempt to elucidate its mechanism of action; compound **1** seems to act predominantly on the cellular membrane of *C. albicans* ATCC 10231, altering its structural integrity, without binding to ergosterol, while inhibiting two important virulence factors—dimorphic transition and biofilm formation—frequently associated with *C. albican**s* pathogenicity and resistance. In conclusion, the present work proved the usefulness of thioxanthones in antifungal therapy as new models for antifungal agents.

## 1. Introduction

The worldwide incidence and severity of fungal infections have increased alarmingly in recent decades, representing one of the main causes of mortality and morbidity within infectious diseases [1]. Systemic fungal infections represent a serious threat to public health, especially among immunocompromised patients, and are responsible for about 1.5 million deaths per year [2]. Among yeasts and filamentous fungi, *Candida*, *Cryptococcus*, and *Aspergillus* species are amongst the most frequently associated pathogenic fungi. However, the emergence of fungi, such as *Scedosporium* spp. and *Fusarium* spp. as etiological agents, has become an increasingly worrying development worldwide [1]. On the other hand, superficial fungal infections have a higher incidence and recurrence than systemic infections, affecting around 20 to 30% of the healthy human population [3]. This type of infection occurs mainly in the skin, hair, and nails, being caused by yeasts (*Candida* spp.) and predominantly by dermatophytes (filamentous fungi of the genera *Trichophyton*, *Microsporum*, *Arthroderma, Nannizzia*, and *Epidermophyton*) [4]. Although most infections are usually superficial, they can become invasive under the conditions of a weakened immune system [5].

The introduction of antifungal agents in clinical practice had an enormous impact on the treatment of fungal infections and their evolution. Currently, antifungals can be divided into five main classes: polyenes, azoles, echinocandins, flucytosine, and allylamines, according to their cellular target and mechanism of action; all these classes of antifungals are commonly used for systemic and/or superficial infections treatment [2]. However, the excessive and inappropriate use of antifungals, as well as the acquisition of resistance genes in fungi, contributed to the increase of resistance that culminated in a situation where treatment of fungal infections has become practically impossible contributing, consequently, to the high mortality and morbidity rates [6]. The extent of antifungal resistance differs by antifungal class: in polyenes, allylamines, and echinocandins it is infrequent, unlike flucytosine and azoles [7]. Given the significant increase in resistant and multi-resistant strains, there is an urgent clinical need to discover and develop new antifungal agents capable of modulating and/or eradicating antifungal resistance [8].

Historically, small synthetic molecules and natural products represent an undisputed source for the discovery of new compounds with antifungal activity [2]. Thioxanthones (9*H*-thioxanthen-9-ones) are *S*-heterocycles with a dibenzo-γ-thiopyrone structure that have gathered great interest in Medicinal Chemistry due to their pharmacological importance [9,10], considered in the 1970’s drugs of choice as antischistosomal agents. In the last decades, thioxanthone derivatives have been widely synthesized and a wide range of biological properties have been studied, leading to the discovery of new agents and the disclosure of their mechanisms of action [10]. Studies carried out by our research group have proven the usefulness of thioxanthones as chemotherapic agents [10], with antibacterial [11,12], antitumor, and efflux pump inhibitory effects [11]. Considering that thioxanthones are bioisosteres of xanthones, another important class with clinical value against fungal infections [13,14], the prospection of new antifungals with a thioxanthone scaffold could be considered a promising strategy to renew the limited therapeutic arsenal and control the emergence of new resistances.

Therefore, in the present work, a library of ten aminothioxanthones (**1**–**10**), including three new derivatives (**8**, **9**, and **10**) synthetized for structure-activity relationship (SAR) purposes, was screened as potential antifungal agents against three fungal strains—*Candida albicans*, *Aspergillus fumigatus*, and *Trichophyton rubrum*—using the Clinical and Laboratory Standard Institute (CLSI) broth microdilution method. For the compounds that showed antifungal activity, the fungal spectrum was further extended to other clinically relevant reference strains and clinical isolates of yeasts and filamentous fungi (including dermatophytes), such as strains intrinsically resistant to fluconazole or with acquired resistance to this azole; these included several species of the genus *Candida* and *Aspergillus*, and (re)emerging species such as *Fusarium* spp. and *Scedosporium* spp. With the small series of compounds available, some SARs of antifungal thioxanthones were established. Furthermore, studies on the effect of the most active compound on two important virulence factors of *C. albicans*—dimorphic transition and biofilm formation—and on mitochondrial activity and cell membrane of *C. albicans* were performed, to elucidate the possible mechanism of action underlying its activity.

## 2. Results and Discussion

### 2.1. Chemistry

Previous results from our research group have demonstrated a set of 1-aminated thioxanthones and structurally related tetracyclic thioxanthenes with potential as antibacterial agents, efflux pump modulators, and/or antibiotic adjuvants. Naturally, antifungal screening of these derivatives (**1**–**7**, Figure 1) was considered to be a suitable starting point for the discovery of potential new antifungal agents; compounds **1**–**7** (Figure 1) were synthesized by copper-catalysed C-N cross-coupling from the commercial 1-chloro-4-propoxy-9*H*-thioxanthen-9-one (**11**) [11]. The in vitro antifungal activity of the seven compounds was evaluated for the first time allowing the identification of a hit compound, 1-[2-(diethylamino)ethyl]-amino-4-propoxy-9*H*-thioxanthen-9-one (**1**). This compound had already proven its usefulness as a cell growth inhibitor, as well as antibacterial activity against reference strains of *Staphylococcus aureus* and *Enterococcus faecalis*, a methicillin-resistant strain of *S. aureus* (MRSA), and synergy with ampicillin and oxacillin against MRSA [11,15]. Moreover, compound **1** had also shown potential at reversing multidrug resistance, as it could inhibit bacterial efflux pumps in an *S. aureus* strain and the formation of biofilm in the same strain [11].

Considering that compound **1** demonstrated such promise as an anti-infective agent, new aminothioxanthones (**8**, **9**, and **10**) were designed and synthesized based on a SAR approach of compound **1** (Figure 1). Additionally, 1-{[2-(diethylamino)ethyl]amino}-4-hydroxy-9*H*-thioxanthen-9-one (**8**) was previously prepared from *p*-chlorophenol and thiosalicylic acid in a multistep way due to difficulties encountered in demethylation [16]; herein a more convenient synthesis is described: compound **8** was obtained smoothly by a demethylation reaction of **1** with a boron tribromide (BBr_3_) agent (procedure b, Figure 1), as described by Palmeira et al. (2012) [15]. To obtain compound **9**, the first step was the same demethylation reaction, following a similar procedure [15], applied to the commercial halogenated thioxanthone **11** (procedure b), which allowed to obtain the product 1-chloro-4-hydroxy-9*H*-thioxanthen-9-one (**12**). From compound **12**, another 1-chloro derivative (1-chloro-9-oxo-9*H*-thioxanthen-4-yl trifluoromethanesulfonate, **13**) was synthesized based on the reaction described by Wu et al. (2008) [17] (procedure c), and the following was submitted for coupling with the primary amine *N*,*N*-diethylethane-1,2-diamine to furnish derivative **14** (procedure d). The cross-coupling reaction used for compound **1** [15] was extended to the 4-amine intermediate **14** (procedure a) to ensure the formation of the desired product 1,4-bis{[2-(diethylamino)ethyl]amino}-9*H*-thioxanthen-9-one (**9**). Compound **12** was also the starting point for obtaining the 4-amine product **10**, which was designed as a hybrid compound comprising a thioxanthone nucleus linked to three known antifungal moieties, namely eugenol and 1,2,3-triazole ring [18,19]. This was performed through propargylation of compound **12**, followed by a conventional copper-catalyzed azide-alkyne cycloaddition reaction of the intermediate **15** with 4-allyl-2-azido-6-methoxyphenol, which was synthetized as described elsewhere [20,21]. Finally, the conversion of intermediate **16** to product **10** was achieved by the same cross-coupling reaction used to obtain **1** and **9**.

The elucidation of the structures of the new aminothioxanthone derivatives, compounds **8**, **9** and **10**, was established by infrared spectroscopy (IR), nuclear magnetic resonance (NMR), high-resolution mass spectrometry (HRMS) (Appendix A) and, in the case of intermediate **16**, X-ray crystallography (Appendix A). Whenever needed, the ^13^C NMR assignments were confirmed by bidimensional heteronuclear single quantum correlation (2D HSQC) and heteronuclear multiple bond correlation (HMBC). The data for derivatives **1**–**7** has been previously shown in [11,15]. As expected, compound **9** showed characteristic data of the structure of thioxanthone, with profiles identical to those described for compound **1** [15]. The proton nuclear magnetic resonance (^1^H NMR) spectra evidenced the presence of signals corresponding to aromatic hydrogens and aliphatic hydrogens present at the C-1 position. In addition, the aliphatic protons of the C-4 position of the newly synthesized compound are evidenced by the higher integration of the two low chemical shift signals (δ_H-1″_ = 1.94–1.82 ppm and δ_H-2″_ = 1.12 ppm). The carbon nuclear magnetic resonance (^13^C NMR) spectra revealed the presence of a highly unprotected signal corresponding to the carbon of the carbonyl group (δ_C-9_ = 183.6 ppm). Furthermore, the signals corresponding to the carbons attached to the amino chains (δ_C-1_ = 143.3 ppm and δ_C-4_ = 147.8 ppm), as well as the signals corresponding to the remaining ten carbons of the thioxanthone ring (δ_C_ = 136.9–106.8 ppm) were also observed. The signals corresponding to the aliphatic carbons showed the lowest chemical shifts, with the methyl group having the lowest value (δ_C-2″_ = 10.7 ppm). As for the characterization of product **10**, it was possible to observe, in addition to the typical signs of the thioxanthone nucleus and amino side chain, the characteristic proton signs of the eugenol subunit, such as those from the allylic chain (δ_H-18_ = 3.37, δ_H-19_ = 5.99–5.89 ppm and δ_H-20_ = 5.14–5.08 ppm) and the two aromatic protons (δ_H-15_ = 6.75 ppm and δ_H-17_ = 7.19 ppm). Furthermore, the signals referring to the proton of the triazole ring (δ_H-11_ = 8.32 ppm) and the methylene group directly attached to this ring (δ_H-9_ = 5.36 ppm) are clearly noted. Apart from the signs related to the thioxanthone moiety, the analysis of the ^13^C NMR spectrum of product **10** showed the expected signals for the carbons of the amino chain (δ_C-6_ = 51.4 ppm, δ_C-7_ = 47.2 ppm, δ_C-5_ = 38.8 ppm and δ_C-8_ = 11 ppm), the hydrogenated carbon the triazole (δ_C-11_ = 123.8 ppm) and those that prove the presence of the eugenol subunit, such as the carbons of the allylic chain (δ_C-19_ = 136.4 ppm, δ_C-20_ = 116.6 ppm and δ_C-18_ = 40.1 ppm), methoxy group (δ_C-21_ = 56.4 ppm) and aromatic hydrogenated carbons (δ_C-17_ = 114.6 ppm and δ_C-15_ = 111.5 ppm).

### 2.2. Antifungal Activity

Aminothioxanthones **1**–**10** and, of note, the chlorinated compound **16** (precursor of **10**) were evaluated for in vitro antifungal activity against two reference strains and a clinical isolate: *Candida albicans* ATCC (American Type Culture Collection) 10231, *Aspergillus fumigatus* ATCC 240305, and *Trichophyton rubrum* FF5, by the CLSI broth microdilution method [22,23]. Compounds **1**, **8**, and **9** exhibited activities against the tested strains (Table 1); no activity was detected for the remaining tested compounds at concentrations up to 128 μg/mL. Given the activity shown by compounds **1**, **8** and **9**, the fungal spectrum was extended to other reference strains and clinical isolates of yeasts (*C. albicans*, *Candida krusei*, *Candida glabrata*, and *Cryptococcus neoformans*) and filamentous fungi (*A. fumigatus* ATCC 240305 and a clinical strains C111, *A. niger* ATCC 16404, *A. flavus* F44, *Fusarium solani* FF125, *F. oxysporum* FF11, *Scedosporium* spp., *Lichtheimia* spp., *Mucor* spp., and dermatophytes *Trichophyton mentagrophytes*, *Nannizzia gypsea*, and *Microsporum canis*). Results of the minimal inhibitory concentration (MIC) and minimal fungicidal concentration (MFC) of the tested compounds and the reference antifungal fluconazole (FL) are shown in Table 1.

In summary, the results obtained indicate the high antifungal potential of compounds **1**, **8**, and **9** against most tested strains of yeast and filamentous fungi (including dermatophytes). Compound **1** was more active against *C. neoformans* CECT 1078, *Scedosporium* spp., *Lichtheimia* spp., *Mucor* spp., and dermatophytes (*T. rubrum* FF5, *T. mentagrophytes* FF7, and *N. gypsea* FF3), with a fungicidal effect achieved for the MIC value or a dilution above (8–32 μg/mL). On the other hand, in *C. krusei* ATCC 6258, a species intrinsically resistant to FL, although the MIC was higher than the previously mentioned strains/species, a fungicidal effect was observed for the MIC value (32 μg/mL). Among the tested strains of *C. albicans* and *C. glabrata*, no substantial difference in susceptibility was observed (either for MIC and MFC), regardless of their susceptibility to FL; FL-resistant strains even showed greater or at least equal sensitivity to FL-susceptible strains. Furthermore, the tested compound exhibited activity against (re)emerging pathogenic fungi *F. solani* FF125, *F. oxysporum* FF115, and *Scedosporium* spp., whose therapeutic options are limited. Finally, and like FL, a fungistatic effect was observed for *A. fumigatus* ATCC 240305 and C111, and *A. niger* ATCC 16404 where sensitivity was equal with a MIC value of 32 μg/mL. In general, the 4-substituted derivatives (**8** and **9**) showed similar activities with compound **1**, since compound **9** is less active than compound **8**. Compounds **8** and **9** showed variable inhibitory effects on the *Candida* strains tested; interestingly, the most susceptible strains were *C. albicans* FL-resistant and *C. glabrata* (including FL-resistant strains). In addition, compound **9** exhibited (unexpectedly) fungicidal activity against azoles-resistant *A. fumigatus* C111, with an MFC value equal to MIC. That said, the three tested compounds (**1**, **8**, and **9**) may be of potential interest as an alternative to the conventional treatment of systemic and superficial fungal infections, especially infections caused by FL-resistant and (re)emerging fungi.

Regarding SAR analysis and comparing the results of 4-propoxy derivatives (compounds **1**–**7**) it was found that the presence of an alkylamine moiety at the C-1 position (compound **1**) appears to be crucial for antifungal activity. Moreover, changes were noted in the alterations of the substituent groups at the C-4 position (Table 1). That is, the presence of a hydroxyl group (compound **8**) or a primary amine (compound **9**) rather than a propoxy group (compound **1**) did not lead to a loss of activity; the two 4-substituted derivatives generally inhibited the growth of almost all the tested fungi at a concentration equal (compound **8**) to or 2x higher (compound **9**) than that observed for compound **1**. The slight structural difference between compounds **8** and **9** can be related to their differences in the demonstrated activity. The presence of a hydroxyl functional group on compound **8** could affect the pKa and, subsequently, its solubility and additionally the possibility of a donor H bond interaction possible with this functional group. Although these preliminary SARs could be established, more derivatives should be synthesized to carry out a more in-depth and complete study on the structure-antifungal activity of thioxanthones, as they have not been described so far, to the best of our knowledge.

### 2.3. Study of C. albicans Phenotypic Virulence Factors

In the *Candida* genus, *C. albicans* is the main pathogenic responsible for both systemic and superficial infections. One of the main characteristics of *C. albicans* pathogenicity is its ability to change morphologically between yeast and hyphae forms and to produce biofilms [24]. In fact, most invasive infections caused by *C. albicans* are associated with biofilm formation on host tissues and medical devices [25]. Furthermore, biofilm formation is one of the mechanisms of resistance to antifungal agents [26]. Thus, there is a pressing need for the discovery and development of new compounds with anti-biofilm activity [27]. Considering the remarkable activity of compound **1** against the tested *C. albicans* strains (including FL-resistant), its influence was evaluated on the two virulence factors, dimorphic transition, and biofilm formation.

#### 2.3.1. Effect on Germ Tube Formation

To evaluate the effect of compound **1** on the dimorphic transition, two strains of *C. albicans* were used: a reference strain, ATCC 10231, and a clinical isolate resistant to FL—H37 (Figure 2). At the concentrations of 2MIC, MIC, and 1/2MIC, complete or almost complete inhibition of germ tube formation occurred, compared to the control. However, for 1/4MIC and 1/8MIC a slight difference is seen between the strains. That is, in ATCC 10231 no significant inhibition was observed, unlike in H37.

#### 2.3.2. Effect on Biofilm Formation

Given the influence of compound **1** on the dimorphic transition, their effect on biofilm formation was evaluated (Figure 3) through the crystal violet assay, which allows for the measurement of total biomass [28,29]. In general, biofilms were significantly inhibited at all concentrations tested, except for the 1/16MIC and 1/32 MIC concentrations on the ATCC 10231 strain (Figure 3). Interestingly, the effect of the tested compound was quite different for the two *C. albicans* strains at 1/8MIC, 1/16MIC, and 1/32MIC concentrations; a greater inhibition of biofilm formation at subinhibitory concentrations was found for the FL-resistant clinical isolate compared to that observed for the reference and sensitive one. That is, the reported percentage of biomass was less than 40% when strain H37 was treated with all tested concentrations of the compound, while for ATCC 10231, higher concentrations were required to cause a comparable effect. Different patterns of susceptibility have been observed for different species and strains of *Candida* for antifungal azoles [30].

In contrast to *Candida* species, the development and behaviour of biofilms from *Aspergillus* and *Trichophyton* species is a poorly understood phenomenon [31,32]. So far, investigations have revealed that filamentous fungi are capable of forming biofilms and that they have similar characteristics to the biofilms formed by *Candida* spp. [33]. Despite the limited knowledge, the demand for new compounds with anti-biofilm activity in filamentous fungi has increased exponentially due to the current clinical problems associated namely with *A. fumigatus* (particularly among immunocompromised patients) and *T. rubrum* [31,32]. That said, the effect of compound **1** on biofilms formed by *A. fumigatus* ATCC 240305 and *T. rubrum* FF5 was also evaluated (Figure 4). At MIC, compound **1** completely inhibited biofilm formation by *A. fumigatus* and *T. rubrum*. Although at 1/2MIC concentration a significant reduction in biofilm formation was also observed for both strains, for *T. rubrum* the effect was much more evident. At 1/4MIC, 1/8MIC, and 1/16MIC concentrations, the effect of the tested compound was quite different for the two strains; a significant inhibition of biofilm formation at subinhibitory concentrations was found for *T. rubrum*, contrary to what was observed for *A. fumigatus*.

To date and to the best of our knowledge, the anti-biofilm activities of thioxanthones have not been reported for these filamentous fungi species. From the analysis of the results obtained (Figure 3 and Figure 4) it was verified that compound **1** was active as an inhibitor of biofilm formation in the tested strains of *C. albicans*, *A. fumigatus*, and *T. rubrum* at a subinhibitory concentration(s). It is known that biofilm formation is associated with resistance to several antifungals and currently there are few azoles (miconazole), polyenes (liposomal formulations of AmB), and echinocandins that are also effective against fungal biofilms [30]. Thus, a possible synergistic effect between the tested compound and an antifungal widely used in clinical practice, but at the same time shows no anti-biofilm effect on *C. albicans*, such as FL, for the treatment of invasive candidiasis would be another valuable therapeutic strategy [34]. Additionally, compound **1** was shown to be a potent inhibitor of *C. albicans* filamentation at subinhibitory concentrations. These results (Figure 2) were consistent with the observed effect on biofilm formation, as the importance of the ability to morphologically change in *C. albicans* biofilm formation is known [24]. Very recent studies carried out by our research group identified compounds **1** and **5** as inhibitors of the biofilm formation of MRSA [11]. This finding indicates that compound **1** appears to be a dual inhibitor of bacterial and fungal biofilm formation, unlike compound **5** (only active as an inhibitor of bacterial biofilm).

### 2.4. Study of Mechanism of Action

Identifying how a potential antifungal act on the fungal cell is as essential as assessing antifungal activity. Understanding the mechanism of action can provide important information in an attempt to improve the antifungal activity of promising compounds and allow their combination with other existing antifungals in therapy [14]. Moreover, their study constitutes an important strategy to limit the emergence of resistance to commercially available antifungals [35]. Given the few reports of antifungal activity of thioxanthones, their modes of action are still unknown, requiring further investigation and clarification. Thus, compound **1** was subject to further studies, to understand the mechanism of action underlying the displayed anti-*C. albicans* activity.

#### 2.4.1. Effect on Cell Mitochondrial Function

Mitochondria are present in most eukaryotic cells and are where the respiratory chain resides, being responsible for metabolic processes and most of the cellular production of adenosine triphosphate (ATP) [36]. Thus, compounds capable of affecting the mitochondrial respiratory chain can be considered potential growth inhibitors and triggers of cell death. The thiazolyl blue tetrazolium bromide (MTT) colorimetric assay is a commonly used method to establish cell viability through the assessment of mitochondrial function; in active cells, yellow tetrazolium salts are cleaved by mitochondrial dehydrogenase enzymes to form purple formazan products, which can be measured and related to mitochondrial activity [35]. The effect of compound **1** on the mitochondrial function of *C. albicans* ATCC 10231, 3 h after exposure, was evaluated through the MTT reduction assay (Figure 5).

Exposure of *Candida* cells to all concentrations of compound **1** (2MIC–1/8MIC) revealed a significant reduction of more than 50% in mitochondrial dehydrogenase activity when compared to the control. The results obtained indicate that the effect of the tested compound at the highest concentrations was identical to that seen for the positive controls, except for FL. As expected, significant inhibition of respiratory chain function was observed in sodium azide- and AmB-treated cells, and untreated cells incubated at 80 °C; sodium azide is a known inhibitor of the mitochondrial respiratory chain [37], AmB is an antifungal of fungicidal activity [38] and the use of high temperatures is a physical process that damages the cell membranes with reduction of mitochondrial function. In contrast, FL-treated cells showed higher mitochondrial activity than control cells and, as such, its behaviour was as expected because, in addition to interfering with ergosterol biosynthesis, the antifungal agent induces the accumulation of reactive oxygen species (ROS). This finding indicates that compound **1**’s fungistatic activity in this yeast can be explained, at least in part, by enzyme dysfunction.

#### 2.4.2. Effect on Membrane Integrity

Along with the MTT reduction assay, the effect of compound **1** on membrane integrity was evaluated by incorporating two fluorescent dyes, SYTO^®^ 9 and propidium iodide (PI), into treated *C. albicans* ATCC 10231 cells; SYTO^®^ 9 is a green dye that penetrates both living and dead cells, whereas PI only penetrates damaged membranes, i.e., dead cells, where it binds to nucleic acids, emitting red fluorescence [39]. Figure 6 shows the percentage of *C. albicans* cells stained with a mixture of SYTO^®^ 9 and PI detected by fluorescence emission after exposure to different concentrations of compound **1** for five min, relative to the control.

At 2MIC and MIC of compound **1**, the decrease in the percentage of *C. albicans* cells stained with SYTO^®^ 9 and PI mixture is attributed to an increase in the percentage of cells permeable to PI. In fact, the values were significantly higher compared to the control. This finding thus evidences a rupture of the membrane integrity and, consequently, a loss of cell viability at these concentrations; behaviour in agreement with the results obtained in the MTT reduction assay. This effect was identical to that observed for AmB-treated cells and untreated cells incubated at 80 °C. Moreover, at concentrations of 1/2MIC and lower, no significant alteration of cell integrity was detected, as in the case of FL and sodium azide. The results obtained for AmB and FL are in agreement with those expected given their known mechanisms of action; in AmB, the compromise of membrane integrity resulting from binding to ergosterol is proven by the increase of cells stained with PI, in contrast to FL, in which the five-minute exposure is not sufficient for a significant percentage of PI given its inhibitory effect on ergosterol biosynthesis [38]. Once confirmed and corroborated for compound **1**, it is inferred that the tested compound has an influence on membrane integrity at 2MIC and MIC concentrations. 

#### 2.4.3. Effect on Potassium Efflux

Given the short incubation period and high permeability of PI observed in the previously discussed study, the effect of compound **1** on membrane structural integrity at MIC appears to result from direct membrane damage rather than as a secondary consequence of metabolic compromise. This type of behaviour is often associated with channel formation, in which an increase in membrane permeability and, consequently, leakage of small ions such as potassium and sodium occurs [38]. Thus, a more extensive study of the effect of compound **1** on the membrane integrity of *C. albicans* ATCC 102313 was carried out using the measurement of potassium ion leakage, five min after exposure, by flame atomic absorption spectrometry.

As shown in Figure 7, at subinhibitory concentrations of compound **1**, no significant potassium efflux was detected when compared to the control. However, in the presence of 2MIC and MIC concentrations, a significantly higher potassium ion efflux was detected, evidencing a change in membrane permeability; the results observed for the positive controls agree with those expected given their known mechanisms of action (explained above). In fact, the increased leakage of the total free potassium content observed confirms the results obtained from previous assays, i.e., that the antifungal activity of the compound may result from a rupture of the cell membrane and, consequently, a loss of cellular components.

#### 2.4.4. Effect on Membrane Ergosterol

Considering the possible influence of compound **1** on cell membrane integrity and to discover its target, the binding capacity of the compound to ergosterol was investigated. Ergosterol is the main component of the fungal membrane, which is responsible for maintaining cell function and integrity [2]. Indeed, the difference in sterol composition between human and fungal cells is what allows many of the current antifungal treatments to be effective [14]. In this study, the effect of the tested compound was evaluated on two reference strains: *Candida albicans* ATCC 10231 and *Aspergillus fumigatus* ATCC 240305, and compared to the antifungals AmB and FL (as controls). The results obtained are represented in the table below (Table 2).

Briefly, the affinity of the compounds with ergosterol was established by determining the MIC in the presence of exogenous ergosterol at 400 µg/mL in the extracellular medium. An increase in the MIC value indicates a possible binding of the tested compound to ergosterol, in which the compound will rapidly form a complex with the exogenous ergosterol and prevent its interaction with the membrane [40]. As expected, the MIC values of AmB increased, as its interaction with ergosterol is already known [2]. Since the presence of exogenous ergosterol in the growth medium results in a decreased binding of the compound to membrane ergosterol, an increase in the concentration of the antifungal is necessary to ensure this interaction at the membrane [41,42]. For FL, no change in MIC values was observed and, as such, its behaviour was in line with what was expected, since the antifungal is known as an inhibitor of ergosterol biosynthesis [2]. Regarding compound **1**, its MIC values remained unchanged for the strains tested in the medium with and without additional exogenous ergosterol. Thus, the results infer that the tested compound does not act at the level of ergosterol, and therefore has a different target. This finding is corroborated by the greater efficacy of compound **1** on the growth of *C. albicans* H37, a strain without ergosterol [43], compared to *C. albicans* strains containing ergosterol in the cell membrane.

In summary, the preliminary studies of the mechanism of action revealed that compound **1** is a potent inhibitor of mitochondrial activity with the ability to affect the structural integrity of the cellular membrane of *C. albicans* ATCC 10231 at growth inhibition concentrations; however, the tested compound does not act by binding to ergosterol. In addition, docking studies revealed a favourable affinity of the tested compound for the enzyme lanosterol 14-α-demethylase of *C. albicans* and, above all, similar to the known inhibitors of ergosterol biosynthesis, itraconazole and (Appendix A). Considering that some azole antifungals have recently shown antitumor activity (e.g., clotrimazole, ketoconazole, and itraconazole [44], the tested compound may act by a mechanism of action common to itraconazole [45] with a level of selectivity for fungi and Gram-positive bacteria. Other studies performed by our research group demonstrated that compound **1** exhibited in vitro antibacterial activity against Gram-positive bacteria (*Staphylococcus aureus* ATCC 25923, ATCC 29213, and B1, *Enterococcus faecalis* ATCC 29212, and *Bacillus subtilis* ATCC 6633) with a range of MIC values of 32–64 μg/mL [11,12], reduces the in vitro growth of a panel of human tumour cell lines, without affecting the growth of non-tumour cells [15,46,47], and causes abnormal in vitro cellular cholesterol localization of human non-small cell lung cancer (NSCLC) cells [48]. Nevertheless, further investigations are needed to fully elucidate the mode of action of this important aminothioxanthone derivative. These results, together with those obtained in this work, highlight the potential value of the aminothioxanthone derivative as an antimicrobial and antitumor agent. Of note, compound **8** was previously identified for its potential as 3-phosphoinositide dependent protein kinase-1 which enzymatically activates several serine/threonine protein kinases [49] and with orthologs in fungi.

## 3. Materials and Methods

### 3.1. Chemistry

#### 3.1.1. Material and General Methods

All reagents and solvents were purchased from Sigma-Aldrich (St. Louis, MO, USA), Alfa Aesar (Ward Hill, MA, USA), Pronalab (State of Mexico, México), TCI (Tokyo, Japan), Acros Organics (Thermo Fisher Scientific, Geel, Belgium), Fisher Scientific (Thermo Fisher Scientific, Loughborough, UK), Chem-Lab NV (Zedelgem, Belgium), or Honeywell Riedel-de Haën (Seelze, Germany), and no further purification process was implemented. Solvents were evaporated using a rotary evaporator under reduced pressure (Buchi Waterchath B-480, BÜCHI Labortechnik AG, Flawil, Switzerland). Reaction progressions were controlled by thin layer chromatography (TLC) using Merck silica gel 60 (GF_254_) precoated plates (0.2 mm of thickness) with appropriate mobile phases. Compounds were visually detected at 254 and 365 nm, and chemicals were used for the visualization of chromatograms. Purifications of the synthesized compounds were performed by flash column chromatography using silica gel 60 (0.040–0.063 mm, Merck, Darmstadt, Germany) and preparative thin layer chromatography (PTLC) using Merck silica gel 60 (GF_254_) plates. Melting points (m.p.) were measured in a Köfler microscope (Wagner and Munz, Munich, Germany). Infrared (IR) spectra were measured in KBr microplates in a Fourier transform infrared spectroscopy spectrometer Nicolet iS10 from Thermo Scientific (Massachusetts, USA) with Smart OMNI-Transmission accessory (Software OMNIC 8.3). The ^1^H and ^13^C NMR spectra were taken at the University of Aveiro, Department of Chemistry on a Bruker Avance 300 spectrometer (300.13 MHz for ^1^H and 75.47 MHz for ^13^C, Bruker Biosciences Corporation, Billerica, MA, USA) or at the *Centro de Materiais da Universidade do Porto* (CEMUP) on a Bruker Avance III 400 spectrometer (400 MHz) in CDCl_3_ or DMSO-*d*_6_ (Deutero GmbH, Ely, UK) at room temperature. Chemical shifts are expressed in δ (ppm) values relative to tetramethylsilane (TMS) as an internal reference. Coupling constants are reported in hertz (Hz). ^13^C NMR assignments were made by 1D, 2D HSQC and HMBC NMR experiments (long-range C, H coupling constants were optimized to 7 Hz), or by comparison with the assignments of similar molecules. The following compounds were synthesized and purified by the described procedures.

#### 3.1.2. Synthesis of 1-Aminated-4-propoxy-9*H*-thioxanthen-9-one (**1**–**7**)

Compounds **1**–**7** had been synthesized and characterized by Durães et al. (2021) [11].

#### 3.1.3. Synthesis of 1-{[2-(Diethylamino)ethyl]amino}-4-hydroxy-9*H*-thioxanthen-9-one (**8**)

To a solution of 1-[2-(diethylamino)ethyl]-amino-4-propoxy-9*H*-thioxanthen-9-one (**1**, 95.90 mg, 0.25 mmol) in anhydrous CH_2_Cl_2_ (2 mL) was carefully added to a solution of 1M BBr_3_ in CH_2_Cl_2_ (90 μL, 0.50 mmol) under a nitrogen atmosphere at −70 °C. After 12 h at room temperature, methanol was added [15], and the mixture was extracted with water and 1M HCl. The organic layer was dried with anhydrous Na_2_SO_4_ and concentrated under reduced pressure to obtain the crude product. Purification by PTLC (SiO_2_, CH_2_Cl_2_:methanol 9:1) gave the pure product 1-{[2-(diethylamino)ethyl]amino}-4-hydroxy-9*H*-thioxanthen-9-one (**8**, 14.2 mg, 17% yield) as an orange solid. M.p. 50–53 °C; IR υ_max_: 3447, 2963, 2918, 2850, 1616, 1570, 1507, 1465, 1262, 1224, 1072; ^1^H NMR (CDCl_3_): δ = 9.83 (1H, s, -OH), 8.51 (1H, d, *J* = 7.6 Hz, H-8), 7.53 (2H, s, H-5 and H-6), 7.40 (1H, s, H-7), 7.14 (1H, d, *J* = 8.9 Hz, H-3), 6.56 (1H, d, *J* = 8.8 Hz, H-2), 4.00 (2H, t, *J* = 6.0 Hz, H-1′), 2.85 (2H, t, *J* = 6.1 Hz, H-2′), 2.71-2.64 (4H, m, H-1″), 1.11 (6H, s, H-2″); ^13^C NMR (CDCl_3_, 75.47 MHz): δ = 183.3 (C-9), 149.2 (C-4), 142.4 (C-1), 136.7 (C-10a), 131.5 (C-6), 130.3 (C-8a), 129.6 (C-8), 129.2 (C-4a), 125.9 (C-5 and C-7), 120.3 (C-3), 113.6 (C-9a), 106.5 (C-2), 51.6 (C-2′), 47.3 (C-1″), 41.5 (C-1′), 11.6 (C-2″).

#### 3.1.4. Synthesis of 1,4-Bis{[2-(diethylamino)ethyl]amino}-9*H*-thioxanthen-9-one (**9**)

To a solution of 1-chloro-4-propoxy-9*H*-thioxanthen-9-one (**11**, 500 mg, 1.64 mmol) in anhydrous CH_2_Cl_2_ (6 mL) was carefully added to a solution of 1M BBr_3_ in CH_2_Cl_2_ (0.56 mL, 3.28 mmol) under a nitrogen atmosphere at −70 °C. After stirring for 12 h at room temperature, methanol was added [15], and the mixture was extracted with water and brine. The organic layer was dried with anhydrous Na_2_SO_4_ and concentrated under reduced pressure to obtain the crude product 1-chloro-4-hydroxy-9*H*-thioxanthen-9-one (**12**, 329 mg). Anhydride pyridine (1.0 mL, 12.5 mmol) was added gradually to a solution of **12** (329 mg, 1.25 mmol) in anhydrous CH_2_Cl_2_ (8 mL) and, after 10 min, Tf_2_O (0.3 mL, 1.88 mmol) at 0 °C. After continuous stirring at room temperature for 24 h, water was added [17], and the mixture was extracted with water, HCl 1M, and brine. The organic layer was dried with anhydrous Na_2_SO_4_ and concentrated under reduced pressure to obtain the crude product 1-chloro-9-oxo-9*H*-thioxanthen-4-yl trifluoromethanesulfonate (**13**, 300 mg). To a solution of **13** (300 mg, 0.76 mmol) in DMSO (8 mL), *N*,*N*-diethylethane-1,2-diamine (0.5 mL, 3.8 mmol) was gradually added, at 90 °C, and stirred for 12 h [17]. The mixture was extracted with HCl 5% and NaOH 20%. The organic layer was dried with anhydrous Na_2_SO_4_ and concentrated under reduced pressure to obtain the crude product 1-chloro-4-{[2-(diethylamino)ethyl]amino}-9*H*-thioxanthen-9-one (**14**, 40 mg). To a mixture of **14** (40 mg, 0.11 mmol) and *N*,*N*-diethylethane-1,2-diamine (31 μL, 0.22 mmol) dissolved in methanol (25 mL) copper(I) iodide (CuI—21.1 mg, 0.11 mmol) and potassium carbonate (K_2_CO_3_—15.3 mg, 0.11 mmol) were added. The reaction mixture was heated at 100 °C in a closed vessel reactor for 48 h. After completing the reaction, methanol was evaporated, and the residue was dissolved with CH_2_Cl_2_ and extracted with water. The organic layer was dried over anhydrous Na_2_SO_4_ and concentrated under reduced pressure to obtain the crude product. Purification by PTLC (SiO_2_, CH_2_Cl_2_:methanol:triethylamine 8:2:0.1) gave the pure product 1,4-bis{[2-(diethylamino)ethyl]amino}-9*H*-thioxanthen-9-one (**9**, 5.1 mg, 10% yield) as a red solid. M.p. 50–54 °C; IR υ_max_: 3446, 2958, 2916, 2850, 1635, 1576, 1507, 1257, 1224, 1071; ^1^H NMR (CDCl_3_): δ = 8.50 (1H, d, *J* = 8.0 Hz, H-8), 7.55–7.51 (2H, m, H-5 and H-6), 7.40 (1H, ddd, *J* = 8.2, 5.0, and 3.3 Hz, H-7), 7.15 (1H, d, *J* = 9.0 Hz, H-3), 6.58 (1H, d, *J* = 9.0 Hz, H-2), 4.01 (2H, t, *J* = 6.7 Hz, H-1′), 3.77–3.53 (2H, m, H-a), 2.86 (2H, t, *J* = 6.9 Hz, H-2′), 2.69 (2H, d, *J* = 7.6 Hz, H-b), 1.94–1.82 (8H, m, H-1″), 1.12 (12H, td, *J* = 7.3 and 2.7 Hz, H-2″); ^13^C NMR (CDCl_3_, 75.47 MHz): δ = 183.6 (C-9), 147.8 (C-4), 143.3 (C-1), 136.9 (C-10a), 131.8 (C-6), 130.0 (C-8a), 129.7 (C-4a), 129.1 (C-8), 126.0 (C-5 and C-7), 119.7 (C-3), 113.4 (C-9a), 106.8 (C-2), 72.3 (C-2′), 71.2 (C-b), 50.4 (C-1″), 47.0 (C-1′), 41.3 (C-a), 10.7 (C-2″).

#### 3.1.5. Synthesis of 4-((1-(5-Allyl-2-hydroxy-3-methoxyphenyl)-1*H*-1,2,3-triazol-4-yl)methoxy)-1-((2-(diethylamino)ethyl)amino)-9*H*-thioxanthen-9-one (**10**)

Cs_2_CO_3_ (156 mg, 0.48 mmol) was added to a solution of 1-chloro-4-hydroxy-9*H*-thioxanthen-9-one (**12**, 64 mg, 0.24 mmol) in acetone (5 mL) and the suspension was stirred for 30 min at 25 °C under a nitrogen atmosphere. Then, propargyl chloride (0.035 mL, 0.48 mmol) was added dropwise. After stirring at 60 °C for 24 h under the same inert atmosphere, water was added and the mixture was extracted with ethyl acetate. The organic layer was dried with anhydrous Na_2_SO_4_ and concentrated under reduced pressure to obtain the product (**15**, 60 mg) pure enough for the next step. This propargyl intermediate (45 mg, 0.15 mmol) and 4-allyl-2-azido-6-methoxyphenol (31 mg, 0.15 mmol) were dissolved in THF (2 mL). Next, it was added dropwise a solution of CuSO_4_.5H_2_O (37.5 mg, 0.15 mmol) and sodium ascorbate (36 mg, 0.15 mmol) in H_2_O (0.5 mL). After stirring at 25 °C for 2 h, the mixture was extracted with ethyl acetate. Then, the organic layer was dried with anhydrous Na_2_SO_4_ and concentrated under reduced pressure to obtain the crude product. Purification by silica gel column chromatography (hexanes: ethyl acetate 7:3–5:5) afforded the pure product 4-((1-(5-allyl-2-hydroxy-3-methoxyphenyl)-1*H*-1,2,3-triazol-4-yl)methoxy)-1-chloro-9*H*-thioxanthen-9-one (**16**, 40 mg, 52%). To a mixture of **16** (30 mg, 0.06 mmol) and *N*,*N*-diethylethane-1,2-diamine (0.016 mL, 0.12 mmol) dissolved in methanol (25 mL), copper(I) iodide (11.4 mg, 0.06 mmol) and potassium carbonate (8.3 mg, 0.06 mmol) were added. The reaction mixture was heated at 100 °C in a closed vessel reactor for 48 h. After this time, methanol was evaporated and the residue was dissolved with CH_2_Cl_2_ and washed with water. The organic layer was dried over anhydrous Na_2_SO_4_ and concentrated under reduced pressure to obtain the crude product. Following, this crude product was dissolved in a minimum volume of chloroform, and hexane was added until precipitation of the pure product 4-((1-(5-allyl-2-hydroxy-3-methoxyphenyl)-1*H*-1,2,3-triazol-4-yl)methoxy)-1-((2-(diethylamino)ethyl)amino)-9*H*-thioxanthen-9-one (**10**, 9 mg, 26% yield) as a red solid. M.p. 63–65 °C; IR υ_max_: 2959, 2928, 2858, 1737, 1612, 1463, 1510; ^1^H NMR (CDCl_3_): δ = 8.49-8.47 (1H, m, H-8), 8.32 (1H, s, H-17), 7.51–7.50 (2H, m, H-5 and H-6), 7.41-7.36 (1H, m, H-7), 7.29 (1H, d, *J* = 9 Hz, H-3), 7.19 (1H, d, *J* = 1.9 Hz, H-23), 6.75 (1H, s, H-21), 6.57 (1H, d, *J* = 9 Hz, H-2), 5.99-5.89 (1H, m, H-25), 5.36 (2H, s, H-15), 5.14-5.08 (2H, m, H-26), 3.98 (2H, dd, *J* = 5.8 Hz and *J* = 2.4 Hz, H-11), 3.94 (3H, s, H-27), 3.37 (2H, d, *J* = 6.6 Hz, H-24), 2.88 (2H, t, *J* = 7.4 Hz, H-12), 2.73 (4H, q, *J* = 7.2 Hz, H-13), 1.13 (6H, t, *J* = 7.2 Hz, H-14); ^13^C NMR (CDCl_3_, 75.47 MHz): δ = 183.3 (C-9), 149.8 (C-20), 148.5 (C-4), 143.9 (C-1), 141.6 (C-19), 136.9 (C-10a), 136.5 (C-16 and C-25), 131.7 (C-6 and C-22), 129.3 (C-8 and C-8a), 129.0 (C-4a), 126.0 (C-5 and C-7), 123.8 (C-17), 121.8 (C-3), 116.7 (C-26), 114.6 (C-23), 112.9 (C-18 and C-9a), 111.5 (C-21), 106.7 (C-2), 66.9 (C-15), 56.6 (C-27), 51.5 (C-12), 47.4 (C-13), 40.2 (C-11), 38.9 (C-24), 11.2 (C-14).

### 3.2. Microbiology

#### 3.2.1. Compounds Preparation

Stock solutions of the tested compounds and standard antifungal drugs were prepared in DMSO (Sigma-Aldrich, St. Louis, MO, USA), stored at −20 °C, and diluted in a fresh RPMI-1640 medium (Biochrom AG, Berlin, Germany) buffered with 3-(*N*-morpholino) propanesulfonic acid (MOPS—Sigma-Aldrich), henceforth referred to as RPMI, just before the assays. Aminothioxanthones **1**–**10** and precursor **16** were prepared at a concentration of 10 mg/mL. FL (Alfa Aesar, Ward Hill, MA, USA), AmB (Sigma-Aldrich), and voriconazole (VOR—Sigma-Aldrich) were prepared at 25.6 and 6.4 mg/mL, respectively.

#### 3.2.2. Fungal Strains

Twenty-four fungal strains were used, including reference strains and clinical isolates of yeasts and filamentous fungi. Yeast strains included three reference strains (ATCC and CECT) and clinical isolates: *Candida albicans* ATCC 10231 (FL-susceptible, S), *C. albicans* H37 (FL-resistant, R), *C. albicans* FF172 (S), C*. albicans* FF176 (R), *C. albicans* DSY294 (S), *C. albicans* DSY296 (R), *C. krusei* ATCC 6258, *C. glabrata* 10R (R), *C. glabrata* DSY562 (S), *C. glabrata* DSY565 (R), and *Cryptococcus neoformans* var. *neoformans* CECT 1078. *C. albicans* H37 was kindly provided by Cidália Pina Vaz (Faculty of Medicine of the University of Porto, Hospital de S. João, Porto, Portugal), and *C. albicans* DSY294, *C. albicans* DSY296, *C. glabrata* DSY562, and *C. glabrata* DSY565 were kindly provided by D. Sanglard (University of Lausanne, Switzerland). Filamentous fungi included reference strains *Aspergillus fumigatus* ATCC 240305 and *A. niger* ATCC 16404, and clinical strains *A. fumigatus* C111, *A. flavus* F44, *Fusarium solani* FF125, *F. oxysporum* FF115, and dermatophytes *Trichophyton rubrum* FF5, *T. mentagrophytes* FF7, *Nannizzia gypsea* FF3, and *Microsporum canis* FF1, and a species of genera *Scedosporium*, *Lichtheimia* (formerly *Absidia*), and *Mucor*. All microorganisms were kept in Sabouraud dextrose broth (SDB—Bio-Mèrieux, Marcy L’Etoile, France) plus glycerol (20%) at −80 °C. Before each assay, a 24–72 h or 5–7 days sub-culture in Sabouraud dextrose agar (SDA—Bio-Mèrieux, Marcy L’Etoile, France) was prepared to achieve optimal growth conditions and purity.

#### 3.2.3. Antifungal Activity

Minimal inhibitory concentrations were determined by the CLSI broth microdilution method of reference documents M27A-3 [23] for yeasts, and M38-A2 [22] for filamentous fungi and dermatophytes. Briefly, cell or spore suspensions were prepared in RPMI from 24–72 h cultures (yeasts and filamentous fungi) or 5–7 day cultures (dermatophytes). The inoculum of yeasts was adjusted to 0.5–2.5 × 10^3^ colony forming units (CFU)/mL while for filamentous fungi and dermatophytes, the inoculum was adjusted to 0.4–5 × 10^4^ and 1–3 × 10^3^ CFU/mL, respectively. Two-fold serial dilutions of the compounds solved in DMSO were prepared in RPMI within the concentration range of 128–0.03 μg/mL. Sterility, growth, and growth in DMSO controls were also included in each assay. The plates were incubated for 48 h at 37 °C (yeasts and filamentous fungi) or 5–7 days at 25 °C (dermatophytes). MICs for the compounds were considered as the lowest concentration inhibiting 100% growth when compared to compound-free controls. MICs of VOR against *C. krusei* ATCC 6258 and *A. fumigatus* ATCC 240305 were determined as quality control. To evaluate the minimal fungicidal concentrations (MFCs), 20 μL of suspension collected from wells without visible growth were transferred to SDA plates. The MFC was defined as the lowest concentration showing 100% growth inhibition after 48 h at 37 °C (yeasts and filamentous fungi) or 5-7 days at 25 °C (dermatophytes). FL was used as a reference antifungal drug, and the MICs were read, as recommended by the CLSI standard, as 50% inhibition for yeasts and filamentous fungi, and 80% for dermatophytes. All the experiments were repeated independently at least three times. A range of values is presented when different results were obtained.

#### 3.2.4. Germ Tube Inhibition Assay

The effect of compound **1** on the dimorphic transition of *C. albicans* reference strain ATCC 10231 and clinical isolate H37 was evaluated from the preparation of cell suspensions (1.0 ± 0.2 × 10^6^ CFU/mL) in NYP medium [*N*-acetylglucosamine (10^−3^ mol/L—Sigma-Aldrich), Yeast Nitrogen Base (3.35 g/L—Difco, New Jersey, USA), proline (10^−3^ mol/L—Sigma-Aldrich), NaCl (4.5 g/L—Sigma-Aldrich), pH 6.7 ± 0.1]. Suspensions were distributed into glass test tubes, to which an appropriate volume of compound stock solution was added to obtain final concentrations ranging between 2MIC and 1/16MIC. Controls were included with and without 1.28% DMSO. Following a three-hour incubation at 37 °C, cells from each sample were observed, and the percentage of germ tubes was determined. Germ tubes were considered when the germinating tube was at least as long as the blastospore [35,50]. The results are expressed as the mean ± standard deviation (SD) of at least three independent assays.

#### 3.2.5. Biofilm Formation Inhibition Assay

The effect of compound **1** on biofilm formation was evaluated through quantification of total biomass by crystal violet assay in a concentration range of 2MIC–1/32MIC using the 96-well flat bottom plate model.

For *C. albicans* (ATCC 10231 and H37 strains), the compound stock solution was added to cell suspension prepared in RPMI at a final concentration of 1.0 ± 0.2 × 10^6^ CFU/mL. Sterility, growth, and DMSO controls were also included in each assay. After 48 h of incubation at 37 °C, the biofilms were washed twice with phosphate-buffered saline 1× (PBS—NZYTech, Lisboa, Portugal) to remove non-adherent cells and stained with crystal violet 1% (*v*/*v*) (from crystal violet 2%, Química Clínica Aplicada, Amposta, Spain), which was removed after 5 min of contact at room temperature. The wells were then gently washed three times with PBS 1× and left to air dry for at least 30 min. Following, acetic acid 33% (*v*/*v*) (from acetic acid 100%, AppliChem, Darmstadt, Germany) was added to completely solubilize the crystal violet and the absorbance of each sample at 570 nm (A_570_) was read [28,29].

For *A. fumigatus* ATCC 240305 and *T. rubrum* FF5, the biofilm formation was performed based on the method described by Costa-Orlandi et al. (2014) [51] with some modifications. The spore inoculum suspension was prepared at a final concentration of 1.0 ± 0.2 × 10^5^ CFU/mL [33] (*A. fumigatus*) or 1.0 ± 0.2 × 10^6^ CFU/mL (*T. rubrum*) in RPMI. Then, 100 μL of inoculum was added to 96-well plates, which were then incubated without agitation at 37 °C for 3 h for biofilm pre-adhesion. After this time, the supernatant was gently removed from each well, the adhered cells were washed twice with PBS 1×, and 100 μL of RPMI and 100 μL of compound stock solution were added. After 48 h of incubation at 37 °C (*A. fumigatus*) or 92 h of incubation at 25 °C (*T. rubrum*), the biofilms were washed twice with PBS 1× and stained with crystal violet 0.5% (*v*/*v*) for 15 min at room temperature. The wells were washed twice with PBS 1× to remove excess stain and the biofilms were decolourized by adding 95% ethanol (from ethanol 100%, Fisher BioReagents, Pittsburgh, PA, USA) to each well. The ethanol solution was gently homogenized with a pipette until the rest of the crystal violet was completely dissolved. Finally, the solution from each well, as well as the sterility, growth, and DMSO controls were transferred to a new 96-well plate and the A_570_ was read.

The biofilm biomass was quantified by measuring the A_570_ in a microplate reader (Thermo Scientific Multiskan^®^ EX, Thermo Fisher Scientific, Waltham, MA, USA). The background absorbance (culture medium) was subtracted from the absorbance of each sample and the data are presented as a percentage of control. At least three independent assays were performed in triplicate for each experimental condition, with the results expressed as mean ± SD.

#### 3.2.6. Yeast Metabolic Viability

The effect of compound **1** on mitochondrial dehydrogenase activity was evaluated through the MTT assay [43]. Briefly, suspensions of *C. albicans* ATCC 10231 with a final cell density of 0.5–2.5 × 10^3^ CFU/mL prepared in RPMI were incubated for 24 h at 37 °C with shaking and then centrifuged. The pellet obtained was resuspended in 2 mL of RPMI and an aliquot was added to the compound stock solution (in RPMI) at different concentrations (2MIC–1/8MIC). After the incubation of 2 h at 37 °C, the cells were centrifuged and exposed to 500 μL of thiazolyl blue tetrazolium bromide (MTT—Alfa Aesar) solution (0.5 mg/mL in RPMI) for 1 h at 37 °C. The purple formazan products of MTT reduction were solubilized with 300 μL of DMSO and their extent was quantified by measuring absorbance at 570 nm using the microplate reader. Untreated cells incubated for 20 min at 80 °C, 10 mM sodium azide (Merck KGaA, Darmstadt, Germany), and AmB and FL at 8 µg/mL were used as controls. The results are expressed as the mean ± SD of three independent assays.

#### 3.2.7. Live/Dead Assay

The effect of compound **1** on yeast viability was evaluated by the incorporation of the two dye components SYTO^®^ 9 and PI, from LIVE/DEAD^®^ BacLight™ Bacterial Viability Kit (Invitrogen, Thermo Fisher Scientific, Waltham, MA, USA), performed according to Stiefel et al. (2015) [39] with some modifications. Briefly, suspensions of *C. albicans* ATCC 10231 were prepared as described above. After 24 h at 37 °C, cells were centrifuged, the pellet was resuspended in 2 mL of 0.85% NaCl and an aliquot was added to the compound stock solution (in 0.85% NaCl) at different concentrations (2MIC–1/8MIC). The cells were incubated for 5 min at 37 °C before being centrifuged and washed with 0.85% NaCl. The fluorescence intensity was determined after a 15-minute incubation in the dark at room temperature of 100 μL of cell suspension resuspended again in 0.85% NaCl and 100 μL of a mixture of 30 μM PI and 5 μM SYTO^®^ 9 were added per well. Fluorescence intensity was measured with a Synergy HT Multi-Detection Microplate Reader (BioTek^®^) with a 488/20 nm excitation filter (both SYTO^®^ 9 and PI), a 528/20 nm (SYTO^®^ 9), and a 590/35 nm (PI) emission filter. Untreated cells incubated for 20 min at 80 °C, 10 mM sodium azide, and AmB and FL at 8 µg/mL were used as controls. The results are expressed as the mean ± SD of three independent assays.

#### 3.2.8. Potassium Efflux

The effect of compound **1** on the cellular membrane integrity of yeasts was assessed by the amount of potassium ions released from the intracellular fungal medium by flame atomic absorption spectrometry, based on the assay performed by Gucwa et al. (2018) [52]. As explained above, *C. albicans* ATCC 10231 suspension was centrifuged and resuspended with 0.85% NaCl (in Milli-Q water). The cells were treated with the stock solution of the compound (in 0.85% NaCl) in a range of concentrations corresponding to 2MIC–1/8MIC and incubated for 5 min at 37 °C. The samples were centrifuged and the supernatant was transferred to new tubes using a cellulose acetate syringe filter with a 0.22 µm pore size. The ion potassium concentration was measured with an AAnalyst 200 Atomic Absorption Spectrometer (Perkin Elmer, Waltham, MA, USA). Untreated cells incubated for 20 min at 80 °C, 10 mM sodium azide, and AmB and FL at 8 µg/mL were used as controls. The results are expressed as the mean ± SD of at least three independent assays.

#### 3.2.9. Ergosterol Binding Assay

The ergosterol (Sigma-Aldrich) was prepared at the time of test execution, where it was first pulverized and dissolved in DMSO, by the desired concentration and volume. The formed emulsion was then homogenized, heated to enhance the solubility, and diluted with RPMI [41]. The MIC of compound **1** against *C. albicans* ATCC 10231 and *A. fumigatus* ATCC 240305 was determined by using the broth microdilution method according to the guidelines of CLSI, as explained above, in the absence and presence of ergosterol. Briefly, two-fold serial dilutions of the compound solved in DMSO were prepared in RPMI containing or not ergosterol at 400 μg/mL [41,42]. Sterility (RPMI and ergosterol alone) and growth controls (suspension with RPMI and ergosterol) were also included in each assay. The MIC was determined after 48 h of incubation at 37 °C. AmB and FL were used as controls. All the experiments were performed in duplicate, with the results expressed as mean ± SD.

#### 3.2.10. Statistical Analysis

Data were analysed using GraphPad Prism Software (version 9.1.2). One-way analysis of variance (ANOVA) was performed followed by Dunnett’s test. Levels of statistical significance at * *p* < 0.05, ** *p* < 0.01, *** *p* < 0.001 were used.

## 4. Conclusions

The present work demonstrated the great potential of three thioxanthone derivatives as new models for antifungal agents. From the tested library, only compounds **1**, **8**, and **9** showed interesting results against most tested strains of yeasts and filamentous fungi, with compound **1** exhibiting the highest activity. Apart from the fungicidal effect against *C. neoformans*, *Scedosporium* spp., *Mucor* spp., and dermatophytes, the tested compounds were effective against FL-resistant strains. Considering the results obtained in the evaluation of antifungal activity, the presence of a linear amine moiety in the C-1 position of the thioxanthone scaffold appears to be a pharmacological characteristic for activity, while the nature of the substituents in the C-4 position does not seem to influence the ability to inhibit fungal growth. Finally, the preliminary assays performed to elucidate the mechanism of action underlying the activity of compound **1** allowed us to infer that it appears to act predominantly on the cell membrane of *C. albicans* ATCC 10231, altering its structural integrity, without interfering with ergosterol, while inhibiting two important virulence factors—dimorphic transition and biofilm formation—related to the pathogenicity and resistance of *C. albicans*. These studies are the first to demonstrate anti-virulence activity, as well as the action of the tested compound on the membrane of *C. albicans*. However, further investigations will be needed to prove the antifungal mechanism of the aminothioxanthone derivative.

## Data Availability

The data presented in this study are available on request from the corresponding authors.

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
