# Peer review of "Antifungal Activity of a Library of Aminothioxanthones"

_antibiotics, 2022, doi:10.3390/antibiotics11111488_

Round 1
Reviewer 1 Report
The authors synthesize a panel of aminothioxanthones, test them for antifungal activity against a panel of pathogenic fungi, and further characterize biochemical and biological features of a restricted subset with antifungal activity. The manuscript is well written and experimental approaches are well designed, thoroughly performed, and clearly described. The choices for compounds to examine, fungi to target, and activities to assess are all reasonable. The experimental results support the interpretations and conclusions. The study goes beyond a screening survey to elucidate in some detail the mechanism and effects of the antifungal activity. The approach for determining that compound 1 exerts a membrane-damaging effect other than by binding to ergosterol is somewhat limited and indirect since it is based on lack of MIC alteration in the presence of high concentration ergosterol, i.e., testing competitive inhibition rather than direct binding. Still, positive and negative controls are included, and it is clear that amphotericin, the classic ergosterol-binding antifungal, does show an MIC impact from exogenous ergosterol whereas compound 1 does not.
Author Response
Author´s response to Reviewers
- Reviewer #1 comments
(x) English language and style are fine/minor spell check required
Author´s response: We understand and acknowledge the reviewer’s comment and some modifications were introduced to better clarify the language and style of the manuscript.
“The authors synthesize a panel of aminothioxanthones, test them for antifungal activity against a panel of pathogenic fungi, and further characterize biochemical and biological features of a restricted subset with antifungal activity. The manuscript is well written and experimental approaches are well designed, thoroughly performed, and clearly described. The choices for compounds to examine, fungi to target, and activities to assess are all reasonable. The experimental results support the interpretations and conclusions. The study goes beyond a screening survey to elucidate in some detail the mechanism and effects of the antifungal activity. The approach for determining that compound 1 exerts a membrane-damaging effect other than by binding to ergosterol is somewhat limited and indirect since it is based on lack of MIC alteration in the presence of high concentration ergosterol, i.e., testing competitive inhibition rather than direct binding. Still, positive and negative controls are included, and it is clear that amphotericin, the classic ergosterol-binding antifungal, does show an MIC impact from exogenous ergosterol whereas compound 1 does not.”
Author's response: We understand and acknowledge the reviewer´s point of view.
Reviewer 2 Report
This manuscript, ‘Antifungal activity of a library of aminothioxanthones’ is devoted to an important direction. The authors have synthesized new derivatives of aminothioxanthones and studied their potential applications as antifungal agents against various fungal as well as yeast strains. This is a need to find new antifungal agents due to enormous increase of infectious diseases so this is very important to scientific community. The article is very well written and cited but there are some points that author should pay attention to,
I recommend authors to provide, HRMS or elemental analysis of their newly synthesized compounds, 8 – 10. Authors mentioned that they have run HMBC or 2D-HSQC to characterize the synthesized compounds but I could not find this in the supporting information (SI). It will be good if authors add these spectra in the SI. I wonder why authors have supported synthesis of 10 by XRD spectra of intermediate 16 while compound 10 is also a red solid. I will suggest authors to include crystal structure XRD of 10.
In line 215, authors wrote that compound 9 is less active than compound 8. There is only slight difference in between the structure of both compounds, only substituent at 4 position is different. How this structural difference is related to the mode of action of these compounds?
Finally, I suggest authors to include in-detailed mechanistic investigation (mode of action) of their compounds, for example supported by DFT simulations or similar.
I recommend the following decision: Accept with minor revision.
Author Response
“This manuscript, ‘Antifungal activity of a library of aminothioxanthones’ is devoted to an important direction. The authors have synthesized new derivatives of aminothioxanthones and studied their potential applications as antifungal agents against various fungal as well as yeast strains. This is a need to find new antifungal agents due to enormous increase of infectious diseases so this is very important to scientific community. The article is very well written and cited but there are some points that author should pay attention to,”
“I recommend authors to provide, HRMS or elemental analysis of their newly synthesized compounds, 8–10. Authors mentioned that they have run HMBC or 2D-HSQC to characterize the synthesized compounds but I could not find this in the supporting information (SI). It will be good if authors add these spectra in the SI. I wonder why authors have supported synthesis of 10 by XRD spectra of intermediate 16 while compound 10 is also a red solid. I will suggest authors to include crystal structure XRD of 10. “
Author´s response: We understand and acknowledge the reviewer’s suggestion. The structure spectra were achieved by 1H and 13C NMR spectra (as supplied in supporting information) and HRMS was now added. We consider that all compounds are fully characterized with sufficient evidence for composition, structure, and purity. To show more clearly that the 1H and 13C NMR spectra are clean, the spectra expansions and bidimensional spectra (HMBC and 2D-HSQC) have been included in the supplementary material. Crystallization trials were undertaken and often yielded solids that were not the X-ray quality single crystals needed to determine the molecular structure of the compound. Unfortunately, this was the case for compound 10 despite several crystallization trials. On the other hand, suitable crystals of the intermediate compound 16 appeared in the crystallization trials.
“In line 215, authors wrote that compound 9 is less active than compound 8. There is only slight difference in between the structure of both compounds, only substituent at 4 position is different. How this structural difference is related to the mode of action of these compounds?”
Author´s response: We understand and acknowledge the reviewer’s point of view. The slight structural difference between compounds 8 and 9 can be related to small differences in the demonstrated activity. One possibility is that the presence of a hydroxyl functional group on compound 8 affects the pKa and, subsequently, its solubility and additionally a donor H bond interaction could be possible with the introduction of this functional group. This explanation was now highlighted in the revised manuscript (please see page 6, lines 232-235 of the revised manuscript).
“Finally, I suggest authors to include in-detailed mechanistic investigation (mode of action) of their compounds, for example supported by DFT simulations or similar”
Author´s response: We thank the reviewer’s suggestion. Although azole drugs are mainly used as antifungal agents, some have recently shown antitumor activity, such as itraconazole. Among the several mechanisms of action already described, itraconazole has the ability to decrease the viability of endothelial and glioblastoma cells (inducing autophagy) by interrupting cholesterol traffic and reducing plasma membrane levels (Liu et al., 2014). Given these facts, in the revised manuscript we propose that compound 1 acts by an antifungal mechanism common to itraconazole; the docking studies performed and now added to the supplementary material support the proposed idea since the tested compound with dual activity (antimicrobial and antitumor) binds with similar affinity to the commercial inhibitor for the enzyme lanosterol 14-α-demethylase. This in-detailed mode of action was now highlighted in the revised manuscript (please see page 13, lines 457-470 of the revised manuscript).
Reviewer 3 Report
Dear authors
thank you very much for your work in the article Antifungal Activity of a Library of Aminothioxanthones.
It is an interesting topic.
However, one of the problems with possible new antifungals, is that you must compare against classical antifungals, and then, you must apply and contrast completely with FLZ data, this includes their cut-off points.
In the final part, you suggest a possible use as a therapeutic agent, but you do not show data concerning the safety of use in humans.
More details are in the pdf file.
kind regards

Author Response
“Dear authors thank you very much for your work in the article Antifungal Activity of a Library of Aminothioxanthones. It is an interesting topic.”
“However, one of the problems with possible new antifungals, is that you must compare against classical antifungals, and then, you must apply and contrast completely with FLZ data, this includes their cut-off points.”
Author's response: We understand and acknowledge the reviewer’s comment. We do not have cut-off values ​​for our compounds, but only for clinically used antifungals such as Fluconazole or others. Therefore, it is not possible for us to categorize our strains in S or R to our compounds. Even though it is not a very low MIC value, it has the huge advantage of having a lower MIC for isolates that show a MIC of >128 to Fluconazole.
Concerning the following points raised by the reviewer in the PDF file:
- “new fungi?? Scedosporium species have been known to be pathogenic for decades.”
Author's response: We understand the reviewer´s comment. In fact, Scedosporium spp. are not new fungi and, therefore, the sentence was removed in the introduction section of the revised manuscript (please see page 1, line 44 of the revised manuscript).
- “MFC minimal fungicidal concent.”
Author´s response: We understand and acknowledge the reviewer’s observation. The designation change, in order to make it more specific for antifungal activity, was introduced in the revised manuscript (please see page 5 - lines 199 and 201, page 6 - lines 212, 224 and 225, and page 16 - lines 636 and 637 of the revised manuscript).
- “if you use de MIC cutoffs for some strains, as C. albicans i.e. are R to your components....”
Author´s response: As mentioned above, we do not have cut-off values ​​for our compounds; only for clinically antifungal drugs.
- “yes, because they are R to FLZ, not to your compounds, or to VRZ i.e.”
Author´s response: This point was fully addressed and is highlighted in the revised manuscript.
- “please specify the culture medium used”
Author´s response: We thank the reviewer’s suggestion. The specification of the culture medium was introduced in the revised manuscript (please see page 16, lines 217 and 218 of the revised manuscript).
- “ok, but 32, if you apply the FLZ MIC, is R to your compound”
Author´s response: This point was fully addressed and is highlighted in the revised manuscript.
- “do you have any evidence that the use of these compounds is safe in humans? do you have data about cytotoxicity of your compounds? pharmacodynamics?...that is necessary before proposing it as a therapy.”
“In the final part, you suggest a possible use as a therapeutic agent, but you do not show data concerning the safety of use in humans.”
Author´s response: We understand and acknowledge the reviewer’s point of view. Previous results showed that compound 1 caused abnormal cellular cholesterol localization and reduced significantly the growth of the human non-small cell lung cancer (NSCLC) cells line in vitro and of NSCLC xenografts in mice, without presenting toxicity to the animals. We now added this information in the revised manuscript. It is not possible to obtain a 100% correlation between in vivo results in animals and in humans. Therefore, further studies are needed to prove the relevance of these compounds in the therapy of fungal infections.
Reviewer 4 Report
I found the authors have done a good job in laying a foundation in identifying and establishing the compound as an effective anti-fungal. Further work will strengthen the utility of the compound.
Author Response
English language and style:
(x) English language and style are fine/minor spell check required
Author´s response: We understand and acknowledge the reviewer’s comment and some modifications were introduced in order to better clarify the language and style of the manuscript.
“I found the authors have done a good job in laying a foundation in identifying and establishing the compound as an effective anti-fungal. Further work will strengthen the utility of the compound.”
Author's response: We acknowledge the reviewer´s comment. Future work will be done to solidify and extend the information already obtained.
Reviewer 5 Report
Lines 46 and 47: In the classification of mycoses there are 4 types: superficial, subcutaneous, systemic and opportunistic. Therefore, in this paragraph it is not correct to call superficial all superficial and subcutaneous mycoses.
Lines 50 and 51: In the new classification of dermatophytes we should write the genera Nannizzia and Arthroderma, in which some of the species of the genera Microsporum and Trichophyton were relocated.
Lines 57 and 58: Terbinafine is an allylamine that is not only used for superficial mycoses, it can also be used for subcutaneous mycoses (Sporotrichosis, Chromoblastomycosis, Eumycetomas), systemic (Histoplasmosis, Coccidioidomycosis, etc.) and opportunistic (Candidiasis, Aspergillosis, etc.).
Lines 84-85 and 183-184: Specify microdilution methods (EUCAST and CLSI).
Lines 188-191: It is important to specify whether we are talking about A. fumigatus or A. section Fumigati, A. niger or A. section Nigri, A. flavus or A. section Flavi, etc.
Table No. 1, it is no longer necessary to repeat the complete name of the genera of each fungus, since it is unnecessary to do so here.
just place
C. albicans
C. glabrata
C. krusei
As for C. neoformans, I think it is convenient to include the variety if it is neoformans or grubii.
A. fumigatus
F. solani, etc.
Lines 200-201: Dermatophytes are also included within the filamentous fungi, it is different to say non-dermatophyte filamentous fungi.
Section 2.3 Should read: Phenotypic virulence factors.
Line 604: C. neoformans var. _____________________
Line 611: Should read N. gypsea and discard M. gypseum, since Nannizzia is considered a separate genus within the dermatophyte classification.
Author Response
“Lines 46 and 47: In the classification of mycoses there are 4 types: superficial, subcutaneous, systemic and opportunistic. Therefore, in this paragraph it is not correct to call superficial all superficial and subcutaneous mycoses.”
Author´s response: We understand and acknowledge the reviewer’s comment. As some authors consider a difference between strictly superficial and mucocutaneous infections, we put it like this, but not to encompass the subcutaneous ones; however, we agree that it can cause confusion. Therefore, the modification was done in the revised manuscript (please see page 1, line 46 of the revised manuscript).
“Lines 50 and 51: In the new classification of dermatophytes we should write the genera Nannizzia and Arthroderma, in which some of the species of the genera Microsporum and Trichophyton were relocated.”
Author´s response: The reviewer is absolutely right in the observation and we are very grateful for the comment. The modification was done in the revised manuscript (please see page 2, line 55 of the revised manuscript).
“Lines 57 and 58: Terbinafine is an allylamine that is not only used for superficial mycoses, it can also be used for subcutaneous mycoses (Sporotrichosis, Chromoblastomycosis, Eumycetomas), systemic (Histoplasmosis, Coccidioidomycosis, etc.) and opportunistic (Candidiasis, Aspergillosis, etc.).”
Author´s response: We understand and acknowledge the reviewer’s comment. As the main therapeutic indication of terbinafine is in superficial mycoses, hence this indication. However, it is used in other situations and therefore the information has been changed in the revised manuscript (please see page 2, line 63 of the revised manuscript).
“Lines 84-85 and 183-184: Specify microdilution methods (EUCAST and CLSI).”
Author´s response: We thank the reviewer’s suggestion. The specification of the microdilution method was introduced in the revised manuscript (please see page 2 - lines 89 and 90 and page 5 - line 190 of the revised manuscript).
“Lines 188-191: It is important to specify whether we are talking about A. fumigatus or A. section Fumigati, A. niger or A. section Nigri, A. flavus or A. section Flavi, etc.”
Author´s response: We thank the reviewer’s suggestion. The specification of species was included in the revised manuscript (please see page 5 - lines 189 and 190 of the revised manuscript).
“Table No. 1, it is no longer necessary to repeat the complete name of the genera of each fungus, since it is unnecessary to do so here. just place C. albicans C. glabrata C. krusei. As for C. neoformans, I think it is convenient to include the variety if it is neoformans or grubii. A. fumigatus F. solani, etc.”
Author´s response: We understand and acknowledge the reviewer’s suggestion. The idea was just to look up the table without resorting to any other means, and for less familiar readers. That's why we only put one time for each one. Nevertheless, the modifications were done in the revised manuscript (please see page 2, Table Nº 1 of the revised manuscript).
“Lines 200-201: Dermatophytes are also included within the filamentous fungi, it is different to say nondermatophyte filamentous fungi.”
Author´s response: We understand and acknowledge the reviewer's suggestion. We chose to simplify the table.
“Section 2.3 Should read: Phenotypic virulence factors.”
Author´s response: The modification was done in the revised manuscript (please see page 6, line 251 of the revised manuscript).
“Line 604: C. neoformans var. _____________________”
Author´s response: The modification was done in the revised manuscript (please see page 16, line 609 of the revised manuscript).
“Line 611: Should read N. gypsea and discard M. gypseum, since Nannizzia is considered a separate genus within the dermatophyte classification.”
Author´s response: We understand and acknowledge the reviewer’s comment. We chose to put the previous designation in order to lead the reader to this taxonomy change. However, we withdrew as suggested by the reviewer on the revised manuscript (please see page 16, line 616 of the revised manuscript).
Round 2
Reviewer 3 Report
Dear authors, thank you very much for improving the article.
However, continue to insist on the "therapeutic use" of compounds in different mycoses. This is not possible without the necessary background, and evidence, for acceptance as a therapeutic agent.
Also, you read in 2 or 3 opportunities of the "preventive" use of your compounds in fungal diseases, how would you use it preventively?
thank you very much.
